

# Sensitivity of Northeast U.S. surface ozone predictions to the representation of atmospheric chemistry in CRACMMv1.0

Bryan K. Place,[1] William T. Hutzell,[2] K. Wyat Appel,[2] Sara Farrell,[1] Lukas Valin,[2] Benjamin N. Murphy,[2] Karl M. Seltzer,[3] Golam Sarwar,[2] Christine Allen,[4] Ivan R. Piletic,[2] Emma L. D'Ambro,[2] Emily Saunders,[5] Heather Simon,[3] Ana Torres-Vasquez,[1] Jonathan Pleim,[2] Rebecca H. Schwantes,[6] Matthew M. Coggon,[6] Lu Xu,[6,7] William R. Stockwell,[8] and Havala O. T. Pye[2]

[1]Oak Ridge Institute for Science and Engineering (ORISE) Fellow Program at the Office of Research and Development, US Environmental Protection Agency, Research Triangle Park, North Carolina, USA
[2]Office of Research and Development, US Environmental Protection Agency, Research Triangle Park, North Carolina, USA
[3]Office of Air and Radiation, US Environmental Protection Agency, Research Triangle Park, North Carolina, USA
[4]General Dynamics Information Technology, Research Triangle Park, North Carolina, USA
[5]Office of Chemical Safety and Pollution Prevention, US Environmental Protection Agency, Washington D.C, USA
[6]NOAA Chemical Science Laboratory (CSL), Boulder, Colorado, USA
[7]Cooperative Institute for Research in Environmental Science (CIRES), University of Colorado, Boulder, Colorado, USA
[8]University of Texas at El Paso, El Paso, Texas, USA

*Correspondence to*: Havala O. T. Pye (pye.havala@epa.gov)

## Abstract

Chemical mechanisms describe how emissions of gases and particles evolve in the atmosphere and are used within chemical transport models to evaluate past, current, and future air quality. Thus, a chemical mechanism must provide robust and accurate predictions of air pollutants if it is to be considered for use by regulatory bodies. In this work, we provide an initial evaluation of the Community Regional Atmospheric Chemical Multiphase Mechanism (CRACMMv1.0) by assessing CRACMMv1.0 predictions of surface ozone ($O_3$) across the Northeast U.S. during the summer of 2018 within the Community Multiscale Air Quality (CMAQ) modeling system. CRACMMv1.0 $O_3$ predictions of hourly and maximum daily 8-hour average (MDA8) ozone were lower than those estimated by the Regional Atmospheric Chemical Mechanism (RACM2_ae6), which better matched surface network observations in the Northeast US (RACM2_ae6 mean bias of +4.2 ppb for all hours and +4.3 ppb for MDA8; CRACMMv1.0 mean bias of +2.1 ppb for all hours and +2.7 ppb for MDA8). Box model calculations combined with results from CMAQ emission reduction simulations indicated high sensitivity of $O_3$ to compounds with biogenic sources. In addition, these calculations indicated the differences between CRACMMv1.0 and RACM2_ae6 $O_3$ predictions were largely explained by updates to the inorganic rate constants (reflecting the latest assessment values) and by updates to the representation of monoterpene chemistry. Updates to other reactive organic carbon systems between RACM2_ae6 and CRACMMv1.0 also affected ozone predictions and their sensitivity to emissions. Specifically, CRACMMv1.0 benzene, toluene, and xylene chemistry led to efficient $NO_x$ cycling such that CRACMMv1.0 predicted controlling aromatics reduces ozone without rural $O_3$ disbenefits. In contrast, semivolatile to intermediate volatility alkanes introduced in CRACMMv1.0



acted to suppress $O_3$ formation across the regional background through the sequestration of nitrogen oxides ($NO_x$) in organic nitrates. Overall, these analyses showed that the CRACMMv1.0 mechanism within the CMAQ model was able to reasonably simulate ozone concentrations in the Northeast US during the summer of 2018 with similar magnitude and diurnal variation as the current operational Carbon Bond (CB6r3_ae7) and good model performance compared to recent modelling studies in the literature.

## 1 Introduction

Both short-term acute and long-term chronic exposure to elevated surface ozone ($O_3$) concentrations can be detrimental to human and ecosystem health (Bell et al., 2005; Rich et al., 2006; Larrieu et al., 2007; Iriti and Faoro, 2008; Ghosh et al., 2018; U.S. Environmental Protection Agency, 2020). The build-up of $O_3$ in the lower atmosphere also has a noticeable impact on Earth's radiative budget (e.g., Brasseur et al., 1998; Stevenson et al., 2013). As a result, many countries and governments
across the world have enacted legislation to limit surface ozone pollution. In the United States the current national ambient air quality standards (NAAQS) for 8-hour daily maximum ozone (MDA8 $O_3$) is set at 70 parts per billion-by volume (ppb) (Bachmann, 2007; U.S. Environmental Protection Agency, 2015). Despite reductions in emissions of precursor gases that lead to $O_3$ formation, many areas across the U.S. are still in nonattainment of these standards (U.S. Environmental Protection Agency, 2022a). Thus, understanding current $O_3$ pollution mitigation strategies and developing new strategies for the future is
pivotal if air quality standards are to be met.

The chemistry of tropospheric $O_3$ formation is complex and involves the non-linear reactions of nitrogen oxides ($NO_x = NO + NO_2$) with reactive organic carbon (ROC) compounds (Seinfeld and Pandis, 2006; Jacob, 1999; Heald and Kroll, 2020). Similarly, formation of secondary fine particle ($PM_{2.5}$) species such as sulfate, nitrate, and secondary organic aerosol (SOA)
involves complex chemistry in multiple phases and is dependent on concentrations of numerous precursor species and atmospheric oxidants. In total, this chemistry can involve thousands of individual chemical compounds and over ten thousand chemical reactions (Dodge, 2000; Stockwell et al., 2012; Jenkin et al., 2015). Due to these complex interactions as well as the role of meteorological and dry deposition processes on $O_3$ and $PM_{2.5}$ (Seinfeld and Pandis, 2006), regulatory bodies use numerical models to simulate past, current, and future (e.g., under modified emission scenarios) concentrations to inform air
quality management. Rather than simulating the explicit chemistry of every known atmospheric compound and reaction, these models usually employ chemical mechanisms which simplify the atmospheric chemistry into a more limited number of species and reactions in order to capture the most important pathways for forming $O_3$ and $PM_{2.5}$ in a computationally efficient manner (Gery et al., 1989; Carter, 1990; Stockwell et al., 1997). Typically, the chemistry leading to $O_3$ is represented separately from the chemistry leading to $PM_{2.5}$ and SOA formation in chemical transport models (e.g., Pye et al., 2010; Koo et al., 2014).




The Community Multiscale Air Quality (CMAQ) model is a numerical model developed by the United States Environmental Protection Agency (U.S. EPA) to estimate $O_3$, $PM_{2.5}$, and other pollutants, both regionally in the U.S. and in other parts of the world (www.epa.gov/cmaq, U.S. Environmental Protection Agency, 2022b). CMAQ is available online (see code availability) and is distributed publicly with three types of chemical mechanisms: the Regional Atmospheric Chemistry Mechanism
(RACM), Carbon Bond (CB) and SAPRC. These three chemical mechanisms represent ozone chemistry with less than a thousand reactions and up to ~200 species and have been tested on multiple model domains where they show acceptable performance at estimating ambient $O_3$ concentrations (e.g., Sarwar et al., 2008; Yu et al., 2010; Sarwar et al., 2013; Mathur et al., 2017; Appel et al., 2021). Currently, Carbon Bond version 6 (CB6r3 as of CMAQv5.3) is the most common mechanism used by the US EPA for predicting $O_3$ (Appel et al., 2021).


The Community Regional Atmospheric Chemistry Multiphase Mechanism version 1.0 (CRACMMv1.0) (Pye et al., 2022) is a next generation chemical mechanism that was distributed for the first time with the release of CMAQv5.4 in October 2022 (U.S. EPA Office of Research and Development, 2022). CRACMMv1.0 builds on the RACM2 framework (Goliff et al., 2013) and includes new representations of several organic systems, most notably monoterpenes and aromatics, and couples gas-phase
with particle-phase products. In addition, the CRACMMv1.0 mechanism provides a built-in transparent mapping of emissions to mechanism species and was designed to conserve emitted carbon as well as track carbon in products as gases react and evolve. These features were included in CRACMMv1.0 to represent particulate matter formation more accurately while also maintaining the ability to predict $O_3$ concentrations.

The goal of this work is to compare CRACMMv1.0 $O_3$ predictions with the previously well-established RACM2 and CB6r3 chemical mechanisms and understand drivers of differences between CRACMMv1.0 and these mechanisms. Future work will present analyses evaluating CRACMMv1.0 $PM_{2.5}$ predictions. For the comparison presented here we used the CMAQ model and performed simulations at 4 km by 4 km horizontal grid resolution for the Northeast United States (US) domain during summer 2018 (Torres-Vazquez et al., 2022). This domain was chosen specifically because areas in the Northeast US frequently
violate the $O_3$ NAAQS (U.S. Environmental Protection Agency, 2022a). In addition, past field studies such as the Long Island Sound Tropospheric Ozone Study (LISTOS) and future field studies (e.g., AEROMMA, Warneke et al., 2022) have been designed to specifically address the issue of high $O_3$ events in the New York City metropolitan area. Air Quality Service (AQS) observations made during the summer of 2018 were used to aid in the evaluation. Finally, a box model was employed to study the different chemical systems and updates that were driving differences in $O_3$ predictions between RACM2 and
CRACMMv1.0.



## 2 Methods

### 2.1 CMAQ model

CMAQ simulations were performed for the Northeast United States (NE U.S.) domain at 4 km by 4 km horizontal grid resolution with 35 vertical layers from June 1 through August 31, 2018 with May 2 through May 31 as the simulation spin up period. In addition to CRACMMv1.0, simulations were also performed with CB6r3 using the AERO7 aerosol module (CB6r3_ae7) (Appel et al., 2021) and with RACM2 using the AERO6 aerosol module (RACM2_ae6) (Sarwar et al., 2013), both of which are available in standard CMAQv5.3.3 (used here) and v5.4 (latest public release). The major difference between AERO6 and AERO7 is in the representation of monoterpene SOA, with AERO7 producing more monoterpene SOA from photooxidation (Xu et al., 2018) and organic nitrates (Pye et al., 2015) than AERO6. Chemical initial and boundary conditions for the NE US domain were generated from previous nested WRF-CMAQ simulations (12 km) which used CB6r3_ae7 (Torres-Vazquez et al., 2022). The initial and boundary conditions from CB6r3_ae7 were mapped to CRACMMv1.0 and RACM2_ae6. See the CMAQv5.4 code repository for mapping of Carbon Bond-based mechanisms to CRACMMv1.0 for boundary and initial condition purposes. Meteorological files for the simulation were generated offline using the Weather Research Forecasting (WRF version 4.1.2) model as described by Torres-Vazquez et al. (2022) and the files were pre-processed through the Meteorology-Chemistry Interface Processor (MCIP) (Otte and Pleim, 2010) for input to the CMAQ simulations.

### 2.2 Emissions

Anthropogenic emissions were created following the 2016 Version 7.2 North American Emissions Modeling Platform (Torres-Vazquez et al., 2022; U.S. Environmental Protection Agency, 2019) with updates described below. The anthropogenic emissions for CB6r3_ae7 are the same as those for the 4 km domain in the work by Torres-Vazquez et al. (2022) and include year-specific mobile emissions predicted by the Motor Vehicle Emission Simulator (MOVES) model, airport emissions following the 2017 NEI's estimates from the Federal Aviation Administration (FAA) airport model, year-specific wildland fires, monitored electric generating unit (EGU) emissions, year-specific commercial marine vehicle emissions, and emissions from other sectors following the 2016v7.2 modeling platform. Primary organic aerosol in CB6r3_ae7 was considered semivolatile and evaporated POA was allowed to undergo gas-phase reaction with OH following the work of Murphy et al. (2017). The empirical representation of anthropogenic SOA sources (pcSOA, Murphy et al. (2017)) was turned off in all cases. For a more complete description of the anthropogenic emissions employed in the CB6r3_ae7 simulations see the work by Torres-Vazquez et al. (2022). Biogenic emissions for all mechanism simulations were calculated within CMAQ v5.3.3 using the EPA's Biogenic Emission Inventory System (BEIS v3.6.1) (Bash et al., 2016).

CRACMMv1.0 emission inputs build on the same methods as the CB6r3_ae7 inputs with a few additional updates. The total mass and speciation of emissions from volatile chemical products were updated to follow VCPy, a model for predicting volatile chemical product (VCP) emissions (Seltzer et al., 2021). Individual ROC species were mapped to CRACMMv1.0 species as



described by Pye et al. (2022). Primary organic aerosol in CRACMMv1.0 was considered semivolatile with volatility profiles of alkane-like emissions for diesel vehicles, gasoline vehicles, and aircraft (Lu et al., 2020) and slightly oxygenated species

profiles for biomass burning and all other POA sources. For sources without specific volatility profiles, the volatility profile of meat cooking emissions was used to produce a lower bound on evaporation of semivolatile species (Woody et al., 2016; Mohr et al., 2009). Semivolatile POA was implemented using the Detailed Emissions Scaling, Isolation, and Diagnostic (DESID) module in all cases (Murphy et al., 2021). The anthropogenic emissions created for CRACMMv1.0 were also used with slight adjustments for RACM2_ae6 simulations in CMAQ (See supplementary information Table S1 for mappings). For

the RACM2_ae6 simulations, primary organic aerosol (POA) was treated as semivolatile with the same volatility profiles as in the CRACMMv1.0 simulations but with the chemistry of AERO6 (Murphy et al., 2017). Alkane-like semivolatile and intermediate volatility organic compounds (S/IVOCs) emitted in the gas-phase were ignored in RACM2_ae6, and the empirical representation of anthropogenic SOA sources (pcSOA, Murphy et al. (2017)) was turned off in RACM2_ae6 as in CB6r3_ae7.

## 2.3 Air quality network observations

Surface-level network observations of air pollutants made in the northeast US between June and August 2018 were used to evaluate CMAQ model outputs. Hourly measurements of $O_3$ and $NO_x$ were obtained from the AQS database using the available pre-generated files and paired in time and space with model quantities using the Atmospheric Model Evaluation Tool (AMET) (Appel et al., 2011). The observations in AQS were quality assured by the reporting agency (e.g., EPA, States, Tribes), and therefore no additional quality checks of AQS data were done in AMET. In the case of time periods with missing data, those

missing periods were removed from the analysis. In cases where multiple observations were reported for a single site using different parameter occurrence codes (POCs), those observations were treated as individual measurements with the POC number used to distinguish between the different measurements for the same site.

## 2.4 Box modelling in F0AM

The Framework for 0-D Atmospheric Modeling (F0AMv4.2) box model was used as a tool to examine differences in chemistry

between the mechanisms (Wolfe et al., 2016). Chemical species and reactions from the RACM2 and CRACMMv1.0 mechanisms were ported into F0AM from CMAQ-ready mechanism files using a custom MATLAB script (see Code and Data Availability). Photolysis rates in RACM2 and CRACMMv1.0 were matched to existing MCM rates in F0AM and the F0AM default example actinic flux rates were prescribed for all simulations. Three chamber experiments were run by initiating experiments with 10 ppb of either α-pinene, isoprene or benzene under high (5 ppb) and low (0.5 ppb) $NO_x$ conditions at

standard temperature (T = 298K) and pressure (P = 1013 mb). Hydrogen peroxide, set at 200 ppb, was used as the radical OH source (~2 x $10^4$ ppb initial OH), and relative humidity was set at 10% across all simulations. After initiation, each chemical system was allowed to evolve for 24 hours to reach steady state before the simulation was terminated. In addition, to gain insight into the role organic versus inorganic updates played in $O_3$ production in CRACMMv1.0, all three ROC precursors were re-run in simulations using a modified RACM2 mechanism (RACM2_mod) where all inorganic rate constants in RACM2



were updated to match those in CRACMMv1.0. This was needed because the development of CRACMMv1.0 not only incorporated updates to various ROC reaction systems in terms of product yields and chemical fates but also included inorganic rate constant updates (>20 rate constants) to reflect current literature values, which differ from those prescribed in RACM2.

## 3 Ozone predictions

### 3.1 Ozone predictions by mechanism

Figure 1a shows the June through August average surface ozone concentration (averaged for all hours) predicted by the CRACMMv1.0 chemical mechanism across the Northeast U.S. model domain. CRACMMv1.0 average ozone predictions ranged from 16-32 parts per billion-by-volume (ppb) with the highest average ozone predictions occurring over the Great Lakes region, Appalachian Mountain region, and the Atlantic coastline. The higher average $O_3$ predictions (28 – 32 ppb) in the Great Lakes region and around Chesapeake Bay (Figure 1) have been shown to be driven by water-land circulation due to

the difference in daytime PBL heights over cool water (typically < 300 m) compared to much higher PBL heights over land (often 1500-2500 m) (Dye et al., 1995; Lennartson and Schwartz, 2002; Foley et al., 2011; Dreessen et al., 2019; Cleary et al., 2022). In particular, $O_3$ exceedance events around Lake Michigan have been predominantly attributed to the northeasterly transport of $O_3$ and $O_3$ precursors to the lake where photochemical $O_3$ production then becomes intensified under conditions of lower vertical mixing and lower dry deposition (Sillman et al., 1993; Dye et al., 1995; Lennartson and Schwartz, 2002;

Foley et al., 2011; Cleary et al., 2022). These lake effects often lead to regular NAAQS exceedances in the region (Foley et al., 2011). The elevated $O_3$ concentrations predicted for the Appalachian Mountain region have also been shown to be driven primarily from the transport of $O_3$ and other pollutants from nearby urban centers and coal-fired power plants (Aneja et al., 1991; Neufeld et al., 2019). In addition, $O_3$ losses in the region have been measured to be lower at the higher elevation on the mountaintops, which leads to the build-up of $O_3$ during the night (Aneja et al., 1991; Neufeld et al., 2019).




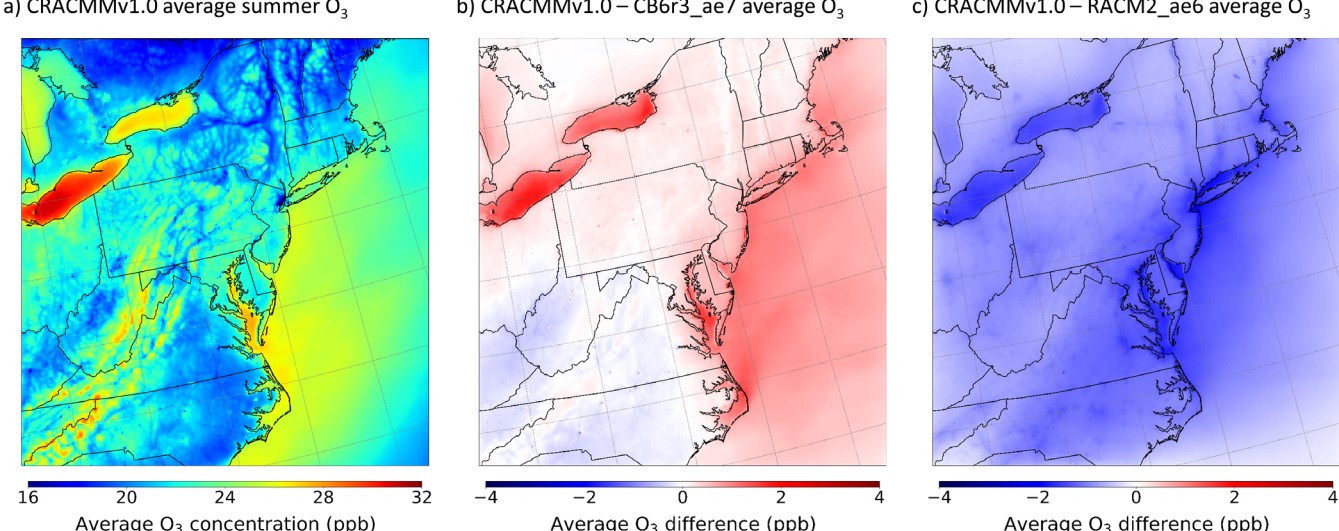

a) CRACMMv1.0 average summer O$_3$       b) CRACMMv1.0 – CB6r3_ae7 average O$_3$       c) CRACMMv1.0 – RACM2_ae6 average O$_3$

**Figure 1. a) Simulated summer (June-August) 2018 surface ozone average (all hours) as predicted by CRACMMv1.0. Simulated summer ozone average (all hours) differences between b) CRACMMv1.0 – CB6r3_ae7 and c) CRACMMv1.0 – RACM2_ae6.**


The magnitude of the ozone concentrations predicted by CRACMMv1.0 were in good agreement with O$_3$ predictions from the base CB6r3_ae7 simulation, with inland differences typically falling below $\pm$ 1 ppb across the model domain (Figure 1b). These absolute differences corresponded to relative differences of $\pm$ 5% (Figure S1a). The largest observed spatial discrepancies between the two mechanisms occurred near bodies of water, where CRACMMv1.0 estimated average ozone

was ~2-4 ppb higher than estimates made by the CB6r3_ae7 chemical mechanism. The higher predicted differences near water are likely explained by intensified chemistry due to the land-water circulation effect described previously, which generally drives the higher O$_3$ concentrations in the regions. In addition, Foley et al. (2011) and Vermeuel et al. (2019) found that O$_3$ production showed greater NO$_x$ sensitivity as urban plumes advected across Lake Michigan. Thus, differences in O$_3$ production near water bodies between the simulations were influenced by the representation of O$_3$-NO$_x$-ROC chemistry in the two

mechanisms and their characterization of the chemical regime. Differences in chemical production of O$_3$ between CRACMMv1.0 and CB6r3_ae7 are discussed and further explored later (Section 4.2). The differences over water between CB6r3_ae7 and CRACMMv1.0 were not expected to be driven by dry deposition over the Great Lakes as deposition is largely supressed over water (Sillman et al., 1993).

Because different VCP emission inventories were employed between the CRACMMv1.0 and CB6r3_ae7 simulations (See Sect 2.2), differences in the two inventory methods, in addition to differences in chemistry, could account for a small fraction of the differences shown in Figure 1b. This would be expected to have the most pronounced effect over urban areas where



VCP emissions are largest. In a previous model study, simulations showed that a complete zero out of VCP emissions led to a 1 ppb $O_3$ change in downtown New York over a 24-hour period (Seltzer et al., 2022); thus, the choice of VCP inventory is
expected to result in differences much less than 1 ppb.

In comparison with RACM2_ae6, CRACMMv1.0 estimated a lower average concentration (average $O_3$ difference of 2-4 ppb) across the model domain, with the largest differences in predictions occurring near urban centers in the metropolitan Northeast in addition to coastal areas along the Great Lakes region and the Atlantic seaboard (Figure 1c). The mechanism-to-mechanism
average $O_3$ differences presented in Figure 1c corresponded to relative average $O_3$ differences of 0 - 15% between the mechanisms across the model domain (Figure S1b). The coupling of meteorology and chemistry, similar to the situation discussed for Lake Michigan, could again explain the larger relative differences in $O_3$ concentrations near water bodies (Figure 1c). Since RACM2_ae6 emissions were mapped from CRACMMv1.0 inputs, the differences between these simulations were due to chemical differences between the mechanisms alone. Over land, differences in $O_3$ predictions between CRACMMv1.0
and RACM2_ae6 were smaller (< 2 ppb, < 7%), but were consistently biased in one direction (Figures 1c, S1b). These findings suggest that updates in chemistry between RACM2_ae6 and CRACMMv1.0 led to a ubiquitous reduction in $O_3$ across the model domain. The role of chemistry as a driver in mechanism-to-mechanism ozone differences between RACM2_ae6 and CRACMMv1.0 is revisited in Section 4.

### 3.2 Evaluation of spatial distribution

Hourly ozone performance statistics were calculated by pairing CMAQ outputs in space and time with 313 AQS sites that reported hourly observations between the months of June and August 2018 using AMET (See Sect 2.3). Figures 2a and 2b show the spatial distribution in model-observation hourly mean biases and linear correlations (r) between predictions and observations for all hourly observations covered by the CRACMMv1.0 simulation. In general, hourly $O_3$ mean biases (MB) indicate a high bias across the model domain, with the highest biases (>15 ppb) occurring along the North Carolina/Tennessee
border (Figure 2a). Model biases were much lower around the metropolitan NE (Washington, D.C., Maryland, New Jersey, New York City/Long Island regions), where predictions fell within ± 4 ppb of the observed average values. Linear correlations between hourly $O_3$ estimates and observations at a given AQS site were typically high (r > 0.8) in the Northeast US (Figure 2b). Correlations between hourly observations and predictions were the weakest at sites located in the Appalachian Mountain region (r = 0.4 – 0.6) and were strongest at sites located along the urban corridor (r > 0.9). Hourly $O_3$ normalized mean biases
(NMB) and normalized mean errors (NME) across the domain can be found in the supplement (Figure S2), and values followed a similar spatial distribution as Figures 2a and 2b with lower NMB (-20% to + 20%) and NME (< 30%) values nearer to urban/sub-urban areas and higher NMB (+ 20-100%) and NME (> 40%) reported for sites located in more rural regions (Figure S2). Even so, hourly ozone predictions had NMB between ±20% at 250 out of 313 sites and NME less than 30% at 227 out of 313 of the reporting sites.




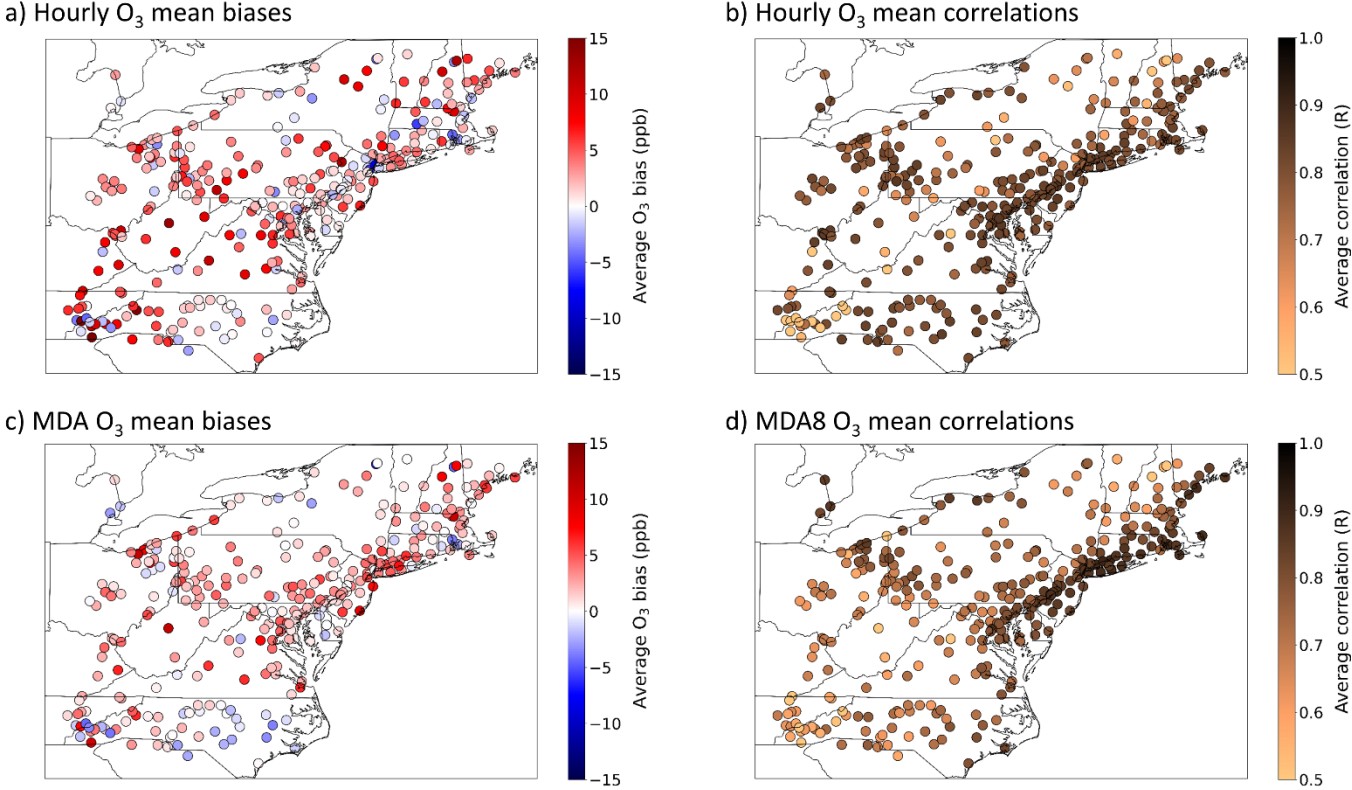

**Figure 2. Ozone (a,c) mean biases (in ppb) and (b,d) correlations between predictions and observations for (a-b) all hourly O₃ values and (c-d) MDA8 O₃ values across the NE US for CRACMMv1.0 calculated using AQS observations between June-August 2018.**

The bias and correlation for daily maximum 8-hour average ozone concentration (MDA8 $O_3$) were also calculated for CRACMMv1.0 at each site (Figures 2c and 2d). Predictions of MDA8 $O_3$ are often used by regulating bodies, such as the US EPA, to determine whether regions are in attainment or nonattainment of national ozone air quality standards. Predictions of MDA8 $O_3$ also reflect a model's ability to estimate daytime $O_3$ concentrations as $O_3$ concentrations are higher during the day. CRACMMv1.0 MDA8 $O_3$ mean biases were similar to the reported hourly $O_3$ biases and ranged from -4 to +16 ppb across the

model domain, with model/observation biases falling within ± 4 ppb at 245 out of 313 sites (Figure 2c). Correlations between modelled and observed MDA8 $O_3$ were also determined to be high (Figure 2d), and CRACMMv1.0 MDA8 $O_3$ predictions showed stronger correlation than hourly $O_3$ predictions at the Appalachian Mountain sites (e.g., Tennessee/North Carolina border) but were weaker in Central North Carolina and in Ohio. MDA8 $O_3$ normalized mean biases did not exceed ±40% with 305 sites reporting normalized mean biases within ± 20% (Figure S2c). MDA8 $O_3$ normalized mean errors did not exceed 45%

across the domain, and NME were lower than 20% for the majority (95%) of sites (Figure S2d).





High-biased hourly and MDA8 $O_3$ predictions were not isolated to the CRACMMv1.0 simulation as both the CB6r3_ae7 and RACM2_ae6 hourly and MDA8 $O_3$ estimates showed high biases over the Northeast in summer 2018 (Figures S3 – S6). High summer $O_3$ daytime and night-time biases have been noted in previous studies in CMAQ investigating air quality over the

Northeast U.S. and CONUS using the RACM2 and CB6 mechanisms (Appel et al., 2021; Sarwar et al., 2013; Cheng et al., 2022). Cheng et al. (2022) noted in their study that daytime high $O_3$ biases were reduced by a more accurate representation of cloud cover via the assimilation of satellite data. Night-time overestimation of $O_3$ in a previous study using CMAQ, on the other hand, was attributed to high $O_3$ coming in from the domain boundaries and low vertical mixing (Li and Rappenglueck, 2018). The exact drivers of the high summer $O_3$ estimates in CMAQ, however, are still under investigation. The calculated

hourly and MDA8 ozone statistics for the CB6r3_ae7 and RACM2_ae6 simulations were found to be of very similar spatial distribution and magnitude to those calculated for CRACMMv1.0 (Figure 2; Figures S2-S6), where both simulations reported lower biases in the metropolitan NE and higher in other areas of the domain. Given that all mechanism $O_3$ biases were lowest nearer to urban areas, this suggests that the CMAQ simulations better estimated $O_3$ concentrations in areas exposed to higher levels of anthropogenic pollutants.


Table 1 summarizes the domain-wide averages of site-specific ozone performance statistics for all three mechanisms and highlights that CRACMMv1.0 performed well when compared with domain-wide hourly and MDA8 $O_3$ estimates from RACM2_ae6 and CB6r3_ae7. The lower $O_3$ estimates by CB6r3_ae7 across the domain most closely matched observations and showed the lowest domain-wide hourly and MDA8 $O_3$ mean biases (MB), normalized mean biases (NMB), and normalized

mean errors (NME). CRACMMv1.0 hourly $O_3$ predictions showed a similar MB (+ 2.7 ppb vs. + 2.4 ppb) and NMB (+ 8.8% vs. +7.9%) to CB6r3_ae7, while CRACMMv1.0 MDA8 $O_3$ MB (+ 2.1ppb vs. +1.5 ppb) and NMB (+7.7% vs. +3.4%) values were slightly higher than CB6r3_ae7. While on average, hourly $O_3$ and MDA8 $O_3$ were slightly overestimated by all mechanisms, the highest $O_3$ values were generally underestimated by all mechanisms (Table 1). For the subset of conditions where observed $O_3$ was above 50 ppb (approximately the highest 10% of concentrations) RACM2_ae6 (MB of -1.7 ppb)

performed best followed by CRACMMv1.0 (MB of -4.7 ppb) and then CB6r3_ae7 (MB of -6.2 ppb). CRACMMv1.0 with the AMORE representation of isoprene chemistry (CRACMM1AMORE) is expected to perform even better than CRACMMv1.0 at high ozone concentrations (Wiser et al., 2022).




**285** **Table 1. Domain-wide site-specific average hourly $O_3$ (number of observations, n=652,476), MDA8 $O_3$ (n=27,037), and hourly $O_3$ above 50 ppb (n=69,103) performance in terms of mean bias (MB), Pearson correlation coefficient (r), normalized mean bias (NMB) and normalized mean error (NME) for the CRACMMv1.0, and CB6r3_ae7, and RACM2_ae6 simulations. The last rows reflect conditions when observed hourly ozone was above 50 ppb.**

| Metric | Mechanism | Domain-wide MB[a] (ppb) | Domain-wide correlation (r) | Domain-wide NMB (%) | Domain-wide NME (%) |
|---|---|---|---|---|---|
| Hourly $O_3$ | CRACMMv1.0 | +2.7 | 0.75 | +8.8 | 27.2 |
| | CB6r3_ae7 | +2.4 | 0.75 | +7.9 | 26.8 |
| | RACM2_ae6 | +4.3 | 0.75 | +14.0 | 28.7 |
| MDA8 $O_3$ | CRACMMv1.0 | +2.1 | 0.76 | +7.7 | 15.8 |
| | CB6r3_ae7 | +1.5 | 0.76 | +3.4 | 13.5 |
| | RACM2_ae6 | +4.2 | 0.75 | +9.6 | 15.9 |
| Hourly $O_3$ above 50 ppb | CRACMMv1.0 | -4.7 | 0.54 | -8.0 | 15.0 |
| | CB6r3_ae7 | -6.2 | 0.53 | -10.6 | 15.2 |
| | RACM2_ae6 | -1.7 | 0.54 | -2.8 | 14.6 |

[a]Equations used for the calculations of MB, r, NMB and NME are reported in the supplement.

**290**

Emery et al. (2017) characterized NMB and NME model statistics from modelling studies reported in the literature (Simon et al., 2012) and found that two thirds of modelling studies reported hourly and MDA8 NMB < 15%, NME < 25% and r > 0.50. With the exception of domain-wide hourly $O_3$ NME, all mechanisms examined here had model performance (NMB, NME and r) within the range of those reported in the literature. By these metrics, CRACMMv1.0 performs consistently with state-of-

**295** science criteria for predicting $O_3$ in photochemical models while also treating the loss of mass to SOA formation.

### 3.3 Evaluation of diurnal distribution

Figure 3a shows the diurnal average hourly ozone surface concentrations ($\pm$ 1 standard deviation) estimated by CRACMMv1.0 (blue trace) compared to average hourly network observations ($\pm$ 1 standard deviation) for all AQS sites (black trace) that reported measurements during the summer of 2018 within the domain. Figure 3a shows that CRACMMv1.0 captured the

**300** general diurnal pattern of the observed ozone concentrations across the model domain and predictions fell within the standard deviation of the observations. CMAQ simulations using CRACMMv1.0 predicted a similar onset in $O_3$ production and an earlier and sharper decline in afternoon $O_3$ than what was typically observed at the AQS sites. The model also predicted a higher average night-time minimum $O_3$ than what was observed. The average summer diurnal $O_3$ concentrations predicted by CMAQ using the CB6r3_ae7 (red dashed trace) and RACM2_ae6 (green dashed trace) mechanisms followed the same diurnal



trend, with CRACMMv1.0 and CB6r3_ae7 simulations showing better agreement with hourly observations than the
       RACM2_ae6 simulation (Figure 3a).

Because the offset observed in morning growth and late afternoon decline in $O_3$ between CMAQ and the AQS observations
was predicted by all mechanism simulations, meteorology was likely a driving contributor to the model-observation
discrepancies during these time periods. For example, a previous study comparing CMAQ $O_3$ predictions across North America
       determined that the timing of the diurnal ozone signal was likely driven by boundary layer dynamics in the model over
       emissions or chemistry (Solazzo et al., 2017). As mentioned in Section 3.2 the high night-time biases observed in Figure 3a
       could have also been driven by meteorology or by $O_3$ coming in from the boundaries (Li and Rappenglueck, 2018). However,
       mechanism-to-mechanism differences, and more specifically, predictions of peak $O_3$ during the daytime, are influenced by the
different treatments of chemistry between the simulations.

To further examine how different treatments of chemistry and/or emissions impacted hourly $O_3$ differences between
mechanisms compared to observations, comparisons at three selected AQS sites (one urban, one suburban and one rural site)
were also plotted in Figures 3b,c,d. Queens, NY was chosen as a representative urban site (Average hourly $[NO_x]_{mod} \approx 12$
ppb), Flax Pond, NY was chosen as a representative suburban site (Average hourly $[NO_x]_{mod} \approx 3$ ppb), and Garrett, MD as a
       representative rural/remote site (Average hourly $[NO_x]_{mod} < 1$ ppb). Similar to Figure 3a, all mechanism predictions fell within
       the standard deviation of the observations at all hours for all three sites (Figures 3b,c,d). The RACM2_ae6 simulation showed
       the greatest diurnal change in hourly $O_3$ concentrations (daytime ozone production) and highest daytime biases while
       CB6r3_ae7 predicted the smallest changes in hourly $O_3$ (daytime ozone production) and showed the lowest daytime biases at
all three sites. All simulations showed the lowest hourly relative biases ($\pm$ 10%) at the urban site (Queens, NY), suggesting
       that the model provides reasonable prediction of $O_3$ production under high $NO_x$ conditions. This reduced bias in an urban area
       is consistent with the hourly $O_3$ biases shown previously across the Northeast (Figures 2; S2-S6), where spatial biases were
       found to be lowest in the metropolitan NE where local ozone formation is expected to make up a larger fraction of total ozone
       than at more rural locations. Larger differences between hourly mechanism-to-mechanism $O_3$ predictions were observed at the
more polluted sites. In particular, the daytime $O_3$ estimated by RACM2_ae6 at Queens and Flax Pond (Figures 3b,c) showed
       a much larger relative increase to CRACMMv1.0 and CB6r3_ae7 than what was seen at Garrett, MD (Figure 3d). Again, this
       may in part be due to larger relative contribution from boundary conditions and transported ozone at rural locations versus
       urban locations. Modelled $NO_x$ concentrations at all the sites were similar between mechanisms (within $\pm$ 0.05 ppb), and the
       relationship between ozone production and $NO_x$ is further explored in the following section.






**Figure 3. Average (± standard deviation) hourly O₃ concentrations predicted by CMAQ using CRACMMv1.0 (blue trace) and observed (black trace) at (a) all AQS sites within the domain; (b) Queens, NY (AQS site 36-081-0124); (c) Flax Pond, NY (AQS site 36-103-0044); and (d) Garrett, MD (AQS site 24-023-0002) during June, July, and August 2018. Predicted average hourly O₃ values in the CB6r3_ae7 CMAQ simulation (dashed red trace) and the RACM2_ae6 CMAQ simulation (dashed green trace) are also overlaid in each panel.**

## 4 Drivers of ozone formation

In this section, CMAQ simulations with emission perturbations are combined with box modelling to understand drivers of ozone formation. In addition, mechanism ozone production efficiency is quantified using modelled NO$_x$ and O$_3$ concentrations across the Northeast US.



## 4.1 Sensitivity to specific ROC emissions

A series of emission sensitivity simulations were performed in CMAQ to gain insight into the precursor ROC systems important for $O_3$ formation in CRACMMv1.0 across the NE US summer 2018 model domain. The sensitivity simulations were 350 conducted by running a set of zero-out simulations (i.e., setting emissions of a chemical class or emissions sector to zero) and determining the response in $O_3$ concentrations to the emission perturbation. A list of all the emission zero out simulations can be found in Table 2. Figure 4 shows domain-wide percent differences in average ozone concentrations ($\Delta O_3$) between the base CRACMMv1.0 simulation and a series of emission zero outs. The largest $\Delta O_3$ response occurred when emissions from biogenic sources were excluded from the simulation (Figure 4a). The biogenic zero-out simulation resulted in percent changes 355 in average $O_3$ concentrations ranging from -10% to +3%. Spatially, average $O_3$ concentrations decreased by ~5-10% in the metropolitan Northeast and increased in the southern part of the model domain in response to the perturbation. Relatively large changes in $\Delta O_3$ were also predicted in the olefin and benzene-toluene-xylene (BTX) zero-out simulations, with average $O_3$ concentration changes ranging from -4% to +2% (Figs 4b and 4c). A similar spatial response in $\Delta O_3$ was seen between the biogenic and anthropogenic olefin zero out simulations (Figs 4a and 4b), while the response of $\Delta O_3$ in the BTX zero-out 360 simulation was localized to urban areas, particularly in the metropolitan NE and never indicated disbenefits (Fig 4c). The chemical formation of $O_3$ in CRACMMv1.0 was less sensitive to large alkanes (HC10) and semivolatile and intermediate volatility organic compound (SVOC+IVOC) emissions across the model domain as a $\Delta O_3$ response of +1% was predicted in these simulations (Figures 4d,e). All five sensitivity simulations showed some reduction in $O_3$ in the New York City urban core with ROC reductions indicating ROC-sensitive ozone formation.


**Table 2: List of emission perturbations relative to the base simulations in CMAQ**

| Chemical mechanism | Emission perturbation |
|---|---|
| CRACMMv1.0 | Benzene, toluene, and xylene-like emissions set to zero |
| CRACMMv1.0 | Biogenic ROC emissions set to zero |
| CRACMMv1.0 | Anthropogenic olefin emissions set to zero |
| CRACMMv1.0 | IVOC (C* range $10^3$ - $10^6$ μg/m$^3$) emissions set to zero |
| CRACMMv1.0 | SVOC (C* range $10^{-2}$ - $10^2$ μg/m$^3$) emissions set to zero |
| CRACMMv1.0 | HC10 (decane and species of similar reactivity) emissions set to zero |
| RACM2_ae6 | Benzene, toluene, and xylene-like emissions set to zero |
| RACM2_ae6 | Biogenic ROC emissions set to zero |
| CB6r3_ae7 | Benzene, toluene, and xylene-like emissions set to zero |



A $\Delta O_3$ response like the one in CRACMMv1.0 was also predicted when biogenic emissions were zeroed out in a simulation
run with RACM2_ae6 (+3% to -10%) (Figure S7), indicating that biogenic emissions were important to $O_3$ formation across
chemical mechanisms in the Northeast U.S. domain. This strong sensitivity of $O_3$ formation to biogenic ROC emissions in the
Eastern and Northeastern United States has also been noted in previous chemical transport model studies (e.g., Hogrefe et al.,
2004; Fiore et al., 2005).  A slightly higher and more widespread decrease in $\Delta O_3$ was seen in the RACM2_ae6 biogenic zero
out simulation (Figure S7) than in the CRACMMv1.0 zero out simulation (Figure 4a), which suggests different representations
of biogenic ROC chemistry between CRACMMv1.0 and RACM2_ae6 lead to some of the differences in modelled $O_3$
concentration shown in Figures 2 and 4. BTX zero-out simulations run using RACM2_ae6 and CB6r3_ae7 (Figures S8 and
S9) resulted in similar $\Delta O_3$ responses (-2% to -4%) around urban areas to those that were observed in the CRACMMv1.0 BTX
zero out simulation (Figure 4c). Domain-wide BTX emission effects on ozone were lower than biogenic emission effects and
more pronounced in urban source regions. Unlike CRACMMv1.0, the RACM2_ae6 and CB6R3_ae7 simulations predicted
slightly higher ozone concentrations ($\Delta O_3$ = +1%) in non-urban regions in the domain in the BTX zero out simulations
compared to the base model run (Figure S8 and S9). Note that the organic nitrate yield in aromatic systems was reduced from
8.2% to 0.2% based on recent work by Xu et al. (2020) in CRACMMv1.0 (Pye et al., 2022). This change increases NO to $NO_2$
conversion which indicates BTX oxidation generally leads to ozone production in CRACMMv1.0. However, CRACMMv1.0
also removes radicals from the gas phase when autoxidation or phenol chemistry leads to SOA thus reducing radical
abundances, and Section 4.2 will illustrate CRACMMv1.0 has a different baseline $O_3$ prediction than RACM2_ae6 for
benzene. These results indicate that the differing representation of aromatic chemical systems within the mechanisms explains
some of the differences in modelled $O_3$ concentrations shown in Section 3.




## a) Biogenic ROC zero out

## b) Olefins zero out

## c) BTX zero out

## d) HC10 zero out

## e) IVOC + SVOC zero out

**Figure 4. Relative changes in O₃ concentrations from the CRACMMv1.0 base simulation (zero out – base) for the a) biogenic emission zero out emission scenario, b) olefin zero out emission scenario, c) BTX zero out emission scenario, d) HC10 zero out emission scenario, and e) IVOC/SVOC zero out emission scenario.**

The modelled reductions in O₃ seen near urban regions (Figures 4a,b,c) and in the New York City urban core specifically (Figures 4a,b,c,d,e) are mechanistically consistent for regions expected to have relatively high emissions of $NO_x$, and thus reductions in ROC would lead to less ozone production. In these more ROC sensitive regions, ozone production drops due to changes in total ROC reactivity. When ROC emission reductions are large enough (such as in the biogenic ROC zero out Fig 4a), even $NO_x$ sensitive locations could transition to a $NO_x$-saturated chemical regime, where ROC reductions reduce ozone.

The emission zero-out simulations often showed less sensitivity in the ΔO₃ response to emission reductions in rural/remote regions (Figures 4a,b,c), and even predicted an increase in O₃ formation in rural regions in response to some emission perturbations (Figures 4a,b,d,e). S/IVOCs and large alkanes (HC10) in particular supressed ozone formation in the base simulation as indicated by their zero out simulations leading to increases in ozone with the exception of the New York City



urban core (Figures 4d,e). The ozone formation potential for HC10 compounds across the entire U.S. for all of 2017 was high
in previous work due to the overall abundance of emissions despite low maximum incremental reactivity (MIR) (Pye et al.,
2022), however a much smaller change in average $O_3$ concentration ($\pm$ 1%) was observed in the HC10 zero-out simulation
here compared to the olefin and BTX simulations. This result suggests that the emissions of HC10 compounds were relatively
less important to ozone formation in the NE US domain compared to the entire US for all of 2017. Given the low MIR of
IVOC and SVOC compounds, zeroing out the emissions of these compounds was expected to have mild impacts on $O_3$
formation, and Figure 4e showed that $O_3$ concentrations increased by ~0.5% across the full domain.

The emission perturbation results suggest that large volatile alkanes (HC10) and SVOC/IVOCs) primarily act to sequester
oxidants such as OH and $NO_x$ thus resulting in increases in $O_3$ for the zero-out simulations. Specifically, S/IVOC alkanes as
well as HC10 in CRACMMv1.0 sequester $NO_x$ with the high efficiency due to a 26-28% yield of alkyl nitrates (Pye et al.,
2022). This hypothesis is supported by observed domain-wide increases (up to 4%) in $NO_2$ when both HC10 and SVOC
emissions are removed from the simulations (Figures S10 and S11). In addition, organic nitrates decrease up to 10% near the
urban core when HC10 emissions are omitted from the simulation (Figure S12). Decreases in organic nitrate formation due to
emission removal could also explain the increases in $O_3$ formation seen in the rural regions of the biogenic and olefin zero-out
simulations (Figures 4a,b), where $O_3$ formation would increase in response to less $NO_x$ loss in a $NO_x$-sensitive regime.

**4.2 Ozone production efficiency**

Ozone production efficiency (OPE) is defined as the number of molecules of $O_3$ produced per molecule of $NO_x$ loss and can
be viewed as a metric describing chain length in $O_3$ propagation before $NO_x$ is chemically removed from the atmosphere
(Jacob, 1999). Thus, model-constrained OPE estimates can provide mechanistic insight into $O_3$-$NO_x$-ROC cycling within a
given chemical system or region. Operationally, OPE has been calculated using the slope of the linear regression between $O_3$
and the sum of all $NO_x$ oxidation products ($NO_z$) as $O_3$ and $NO_z$ evolve during the photochemically active hours of the day
(e.g., Arnold et al., 2003; Sarwar et al., 2013; Henneman et al., 2017). This OPE proxy ($\Delta O_3/ \Delta NO_z$) provides a good first-
order approximation of OPE but may not sufficiently capture ozone recycling in regions impacted by fresh $NO_x$ emissions and
regions where $NO_x$ and $NO_z$ losses through deposition are high. Using this proxy (i.e. $\Delta O_3/ \Delta NO_z$) we estimated mechanism
domain-wide OPE values for the Northeast US (Figure 5). This calculation leveraged the fact that different locations
experienced air masses of different ages and $\Delta O_3/ \Delta NO_z$ can be calculated using the linear relationship between $O_3$ and $NO_z$
concentrations across all grid cells in the model domain for each hour of the day. The OPE proxy showed very strong linear
correlations between $O_3$ and $NO_z$ (r > 0.7) between the hours of 11:00 and 17:00 local time. The $\Delta O_3/ \Delta NO_z$ values showed a
linear increase from the morning to the evening for all three mechanisms and were consistently highest for the RACM2_ae6
simulation, and consistently lowest for the CB6r3_ae7 mechanism for all hours of the day. The OPE values evolved at similar
rates during the day between the three mechanisms and reached a peak between the hours of 16:00 and 17:00 local time (Figure
5).



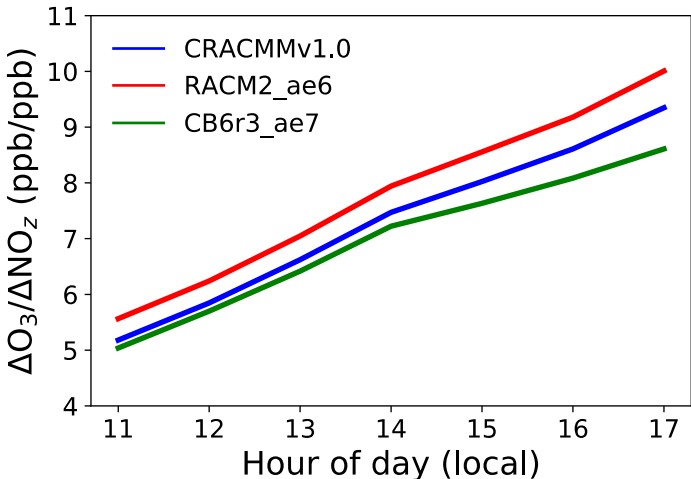

**Figure 5. Average domain-wide hourly ozone production efficiency (OPE) calculated from the slope of the linear regression between NO$_z$ vs O$_3$ at a given hour between 11:00 and 17:00 local time for the CRACMMv1.0, RACM2_ae6, and CB6r3_ae7 mechanism base**
**simulations.**

Figure 5 indicates that there are either differences in O$_3$ production, or NO$_x$ recycling, or a combination of both between mechanisms and that these differences persist at all hours during the day. The trend in OPE values (CB6r3_ae7 < CRACMMv1.0 < RACM2_ae6) is consistent with the diurnal trends in the modelled O$_3$ concentrations observed in Figure 3.

This trend in mechanisms was noted in a previous study model where RACM2_ae6 OPE predictions were shown to be consistently higher than Carbon Bond version 5 (specifically CB05TUCL) OPE predictions leading to a poorer match with observations than Carbon Bond in the Southeast US (Sarwar et al., 2013). Figure 5 confirms that updates between RACM2_ae6 and CRACMMv1.0 led to decreases in OPE and improvement in CRACMM O$_3$ predictions with observations in the Northeast (Figure 2; Table 1). In the following section, differences in the representation of chemical systems between RACM2_ae6 and

CRACMMv1.0 that may have led to differences in ozone production and/or NO$_x$ loss between the two mechanisms are further explored.

**4.3 Box model simulations**

The F0AM box model (Wolfe et al., 2016) was used to further probe the mechanistic drivers of differences between the CRACMMv1.0 and RACM2 chemical mechanisms that could be important for photochemical O$_3$ production. Note for this

study, only the gas-phase aspects of the RACM2 base mechanism from CMAQ were ported and tested in F0AM; thus, RACM2 rather than RACM2_ae6 nomenclature will be used to refer to these results throughout this section. The box model investigation focused on RACM2 and CRACMMv1.0 because the definitions of chemical species and ROC families are similar between mechanisms, allowing for a more direct chemical comparison between the mechanisms. In addition, CRACMMv1.0 was built upon the RACM2 framework and can be more incrementally tested. Differences in chemistry between Carbon Bond-



and RACM-based mechanisms have been explored previously (Sarwar et al., 2013) and detailed analyses are beyond the scope of this study.

Box model simulations were initiated in batch mode with 10 ppb of a precursor ROC, 200 ppb of $H_2O_2$ (OH source), and either 5 ppb of $NO_2$ ($NO_X$ conditions typically observed at the Queens, NY and Flax pond, NY sites from Figure 3) or 0.5 ppb $NO_2$

($NO_X$ conditions typically observed at the Garrett, MD site from Figure 3). The chemical systems were allowed to evolve for 24 hours to reach steady state (See Sect. 2.5 for a full description of the model setup). The dominant fate of $RO_2$ in simulations under high $NO_x$ conditions was confirmed to be $RO_2 + NO$, while simulations initiated with $NO_x$ concentrations of 0.5 ppb were dominated by $RO_2 + RO_2$ reactions. For each simulation, the evolution of $O_3$ was monitored over time. Box model simulations were run with α-pinene, isoprene, and benzene as the ROC precursors because the α-pinene and aromatic chemical

systems underwent major updates in CRACMM compared to RACM2. Additionally, the CRACMMv1.0 and RACM2 biogenic and BTX zero out simulations (Figures 4, S7-S9) showed substantial impact on ambient $O_3$ concentration (anthropogenic olefin chemistry, although important for $O_3$ formation, remained unchanged between RACM2 and CRACMMv1.0).

The production of $O_3$ over time predicted by RACM2 and CRACMMv1.0 under both high and low $NO_x$ conditions is plotted in Figure 6 for all three ROC precursor system simulations. The evolution of $O_3$ over time followed similar trends in both mechanisms and confirms that updates made to the different ROC systems in CRACMMv1.0 did not lead to massive changes in the kinetics of ozone production. For all three high $NO_x$ (5 ppb) simulations, RACM2 led to higher $O_3$ predictions than CRACMMv1.0. The largest mechanism differences in $O_3$ production occurred in the simulation run with α-pinene under higher

$NO_x$ conditions, where 31.1 ppb of $O_3$ was produced by CRAMMv1.0 versus 35.8 ppb produced by RACM2 by the end of the simulation (Figure 6a). The absolute difference in $O_3$ production between RACM2 and CRACMMv1.0 (CRACMMv1.0 – RACM2, -3.2 ppb) in the α-pinene high $NO_x$ simulation corresponded to a relative difference of -13.1% (Table 3). The differences in $O_3$ between CRACCMv1.0 and RACM2 for the simulations run with isoprene (36.8 vs 38.9 ppb of $O_3$) and benzene (33.3 vs 34.2 ppb of $O_3$) under high $NO_X$ conditions were lower than those predicted for α-pinene (Figures 6b,c), but

still indicated mechanism differences of up to -5.7% (Table 3). The total amount of $O_3$ produced in the three simulations under low $NO_x$ conditions (0.5 ppb) was lower and ranged from 4.7 to 9.9 ppb (Figure 6) with the overall changes in ozone between mechanisms very minor for the isoprene and benzene systems ($O_3$ changes within 2.2%). The largest relative changes in $O_3$ production under lower $NO_x$ conditions (-26.3%) between the mechanisms was again observed in the simulation initiated with α-pinene.




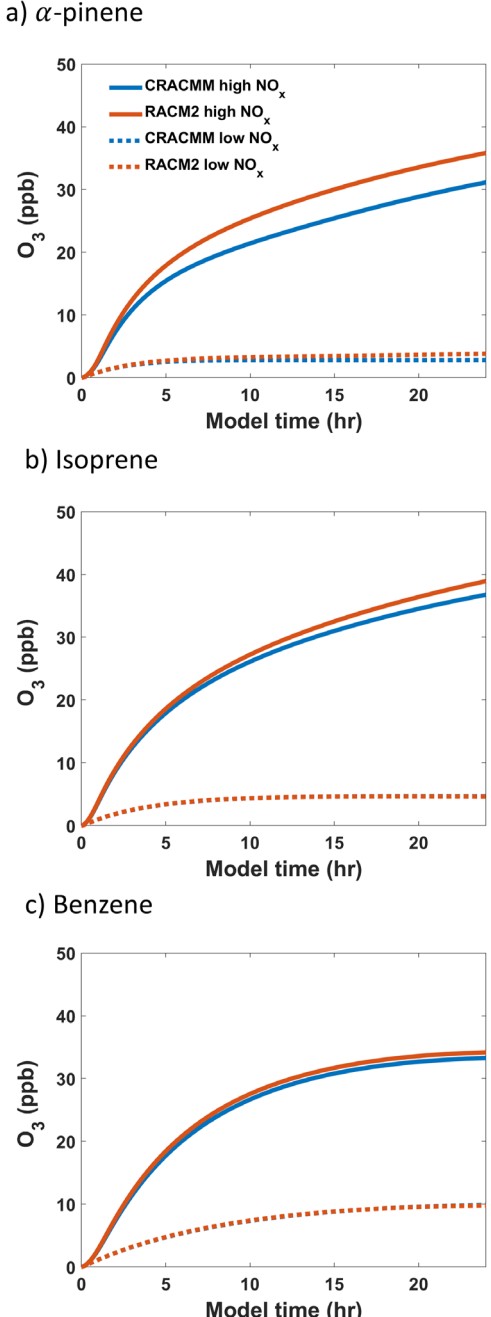

**Figure 6. Evolution of O₃ from photochemical oxidation simulations in the F0AM box model using a) α-pinene and b) isoprene and c) benzene as ROC precursors under high NO$_x$ (5ppb) and low NO$_x$ (0.5 ppb) conditions.**



The absolute and relative differences in O$_3$ production between the two mechanisms were reduced in almost all simulations
      when RACM2 inorganic rates were updated (RACM2_mod) to match those in CRACMMv1.0 (Table 3). The relative
      difference in O$_3$ production in the simulations initiated with 5 ppb NO$_2$ and 10 ppb ROC using RACM2_mod decreased from
      -13.1% to -10.4% in the α-pinene simulation, decreased from -5.7% to -2.1% for the isoprene simulation, and decreased from
      -2.6% to -1.8% in the benzene simulation. Further, in the low NO$_x$ simulations run with RACM2_mod, O$_3$ differences were
reduced to within 0.5% of CRACMMv1.0 for the isoprene and benzene systems. The only simulation that showed an increase
      in O$_3$ production when RACM2_mod was run in place of RACM2 was the simulation run with α-pinene under low NO$_x$
      conditions, where relative differences in O$_3$ production increased from -26.3% to -28.2%.

**Table 3. Absolute and relative differences between CRACMMv1.0 and RACM2 in the amount of ozone produced**
**(ppb) in box model simulations run with α-pinene, isoprene and benzene under both low NO$_x$ (0.5 ppb) and high NO$_x$**
      **(5 ppb) conditions. All results are reported relative to CRACMMv1.0.**

| ROC precursor | Chemical mechanism difference | Absolute difference in O$_3$ (high NO$_x$) | Relative difference in O$_3$ (high NO$_x$) | Absolute difference in O$_3$ (low NO$_x$) | Relative difference in O$_3$ (low NO$_x$) |
|---|---|---|---|---|---|
| α-pinene | CRACMMv1.0 – RACM2 | -4.7 ppb | -13.1% | -1.0 ppb | -26.3% |
| Isoprene | CRACMMv1.0 – RACM2 | -2.1 ppb | -5.7% | +0.1 ppb | +2.2% |
| Benzene | CRACMMv1.0 – RACM2 | -0.9 ppb | -2.6% | +0.1 ppb | +1.0% |
| α-pinene | CRACMMv1.0 – RACM2_mod | -3.6 ppb | -10.4% | -1.1 ppb | -28.2% |
| Isoprene | CRACMMv1.0 – RACM2_mod | -0.8 ppb | -2.1% | <0.1 ppb | <0.5 % |
| Benzene | CRACMMv1.0 – RACM2_mod | -0.6 ppb | -1.8% | <0.1 ppb | <0.5% |

      The results presented in Table 3 indicate that differences in the representation of organic chemistry in CRACMMv1.0 vs.
      RACM2 do partially explain the differences in O$_3$ concentrations from CMAQ across the Northeast US model domain, given
that mechanism differences in O$_3$ production still remained in all simulations after inorganic rate constants were matched
      between the mechanisms. In particular, a majority of the observed O$_3$ differences in the α-pinene-NO$_x$-O$_3$ system ($\geq$ 80%)
      under both high and low NO$_x$ conditions resulted from changes to the organic reactions alone. A high fraction of the O$_3$
      differences (~70%) in the benzene- NO$_x$-O$_3$ system were also driven by organic reaction updates for the simulations run with





higher NO$_x$. As anticipated, organic reaction changes updates played a smaller role in the simulations with isoprene, however
a difference in O$_3$ production of 2% still remained after running the simulations with RACM2_mod. Since RACM2_ae6 O$_3$
predictions in CMAQ were shown to be generally biased high for the Northeast (Table 1) and biogenic emissions were shown
to be important for ozone formation (Figure 4a), reductions in O$_3$ production in CRACMMv1.0 contributed to the more
accurate average O$_3$ predictions across the Northeast US compared to RACM2_ae6. Previous work has found properly
representing monoterpene chemistry, in particular, is important for accurately predicting organic nitrates and thereby ozone
across North America (Browne et al., 2014; Fisher et al., 2016; Zare et al., 2018) including in the Northeast US (Schwantes et
al., 2020).

Further investigation into the mechanisms revealed that there were also differences in the predicted loss of NO$_x$ between
RACM2_mod and CRACMMv1.0 (Figure S13), and that the differences in the evolution of NO$_x$ with time were highest in the
experiment run with α-pinene. Thus, the parameterization of monoterpene reactions (which included the addition of
autoxidation and explicit second-generation chemistry of monoterpene nitrates and aldehydes) led to both decreased O$_3$
production and increased loss of NO$_x$ in CRACMMv1.0 vs RACM2. Despite a reduction in organic nitrate yield in the benzene
system (0.2% in CRACMMv1.0 and 8.2% in RACM2_mod) there was also higher NO$_x$ loss observed in the benzene simulation
run with 5 ppb NO$_2$ (Figure S13). Overall, the mechanism differences in NO$_x$ loss, in addition to ozone production, are
consistent with predicted differences in OPE across the Northeast U.S. in CRACMMv1.0 vs RACM2 (Figure 5).

## 5 Conclusions

This study provides the first evaluation of O$_3$ predictions using the newly developed CRACMMv1.0 chemical mechanism in
the context of other currently available mechanisms and demonstrates CRACMMv1.0 can provide accurate ozone predictions.
Average O$_3$ predictions across CRACMMv1.0, CB6r3_ae7, and RACM2_ae6 simulations during the summer of 2018 over
the Northeast US were generally within ± 10% of each other and all had domain-wide mean biases of less than 5 ppb.
Mechanism differences were most pronounced over bodies of water where meteorology amplified differences. Over land,
domain-wide O$_3$ estimates in CRACMMv1.0 were found to be of similar magnitude to the CMAQv5.3.3.3 operational
mechanism (CB6r3_ae7) (± 1 ppb) but were universally lower in the mechanism upon which CRACMMv1.0 was built
(RACM2_ae6) by 1-3 ppb. The lower O$_3$ concentrations and OPE in the CRACMMv1.0 simulation compared to RACM2_ae6
resulted in better predictions of all-hour and MDA8 O$_3$ concentrations across the NE region as indicated by reductions in the
mean bias, normalized mean bias, and normalized mean error.

CRACMMv1.0 evaluation against AQS ozone observations indicated it is more skilled at predicting ozone in locations with
elevated ozone which is important for understanding sources of exposure at concentrations most likely to cause harm.
CRACMMv1.0 showed improved performance over the current CMAQ operational mechanism (CB6r3_ae7) when hourly



ozone was elevated above 50 ppb. Spatially, CRACMMv1.0 showed smaller bias in the Northeast U.S. urban corridor and higher bias at rural sites, particularly in the Appalachian Mountains. Similar results were found for diurnal predictions at individual sites where CRACMM best matched $O_3$ observations at a site that experienced higher $NO_x$ concentrations. As regional boundary conditions for CRACMMv1.0 were obtained from CB6r3_ae7, the full effects of CRACMMv1.0 on 550 regional background air quality and long range transport predictions have yet to be fully examined.

Improvements in CRACMMv1.0 compared to RACM2_ae6 $O_3$ predictions were driven by updates to the inorganic reaction rate constants as well as updates in the representation of organic chemistry. These updates also caused slight changes in the sensitivity of ozone to ROC precursor emissions. Box modelling simulations in F0AM showed lower $O_3$ production and higher 555 $NO_x$ loss for monoterpene oxidation consistent with the lower overall OPE predicted across the Northeast with CRACMMv1.0 compared to RACM2_ae6. The zero out simulations revealed that domain wide average $O_3$ estimates slightly increased when emissions of S/IVOCs were omitted, suggesting the inclusion of these emissions played a role in $O_3$ formation and mainly acted to reduce ozone. As S/IVOCs are not integrated with radical chemistry leading to ozone in RACM2_ae6 or CB6r3_ae7, some changes in the sensitivity of ozone to emissions are expected in CRACMMv1.0 compared to current mechanisms. As 560 further example, BTX emission zero outs indicated rural ozone is relatively insensitive to aromatic emissions in CRACMMv1.0 whereas RACM2_ae6 (and CB6r3_ae6) predicted ozone dis-benefits (increases) in the rural Northeast when aromatic emissions were removed.

Isoprene and monoterpenes, largely from biogenic sources, are examples of chemical systems where accurate representation 565 of their chemistry across phases is critical to improve prediction of both ozone and fine particle endpoints. As with RACM2_ae6, CRACMMv1.0 $O_3$ concentrations showed great sensitivity to biogenic emissions emphasizing the need to represent their $NO_X$ cycling and radical chemistry well. In addition, autoxidation products with low volatility that sequester radicals are abundant from monoterpenes and critical for SOA formation (Pye et al., 2019). Separate work building on CRACMMv1.0, demonstrated that updated isoprene chemistry led to improved ozone predictions at high (>50 ppb) 570 concentrations as well as predictions of isoprene epoxydiol SOA precursors (Wiser et al., 2022). This need to have gas-phase mechanisms predict intermediates leading to SOA and have SOA products removed from the gas-phase was a major motivation behind the development of CRACMM. Future evaluation of the fine particle predictions of CRACMMv1.0 will provide even further constraints on the radical chemistry leading to ozone explored here.

**Code and data availability**

The implementation of RACM2_ae6 and CB6r3_ae7 used here are available in CMAQ v5.3.3 (U.S. Environmental Protection Agency Office of Research and Development, 2019). CRACMMv1 is available in CMAQ v5.4 (U.S. EPA Office of Research and Development, 2022). Supporting data for CRACMM including guidance on emission preparation is available at



https://github.com/USEPA/CRACMM (U.S. Environmental Protection Agency, 2022c). AMET is available at https://github.com/USEPA/AMET (U.S. Environmental Protection Agency, 2022d). F0AM is available at
580 https://github.com/AirChem/F0AM (Wolfe, 2022). Specific analyses and scripts used in this manuscript, such as the modelled and observed ozone concentrations, F0AM box model inputs, and exact CMAQ code used, will be archived on data.gov with a persistent identifier (doi available in final version).

**Competing interests**

The authors declare that they have no conflict of interest.

**Disclaimer**

The views expressed in this article are those of the authors and do not necessarily represent the views or policies of the U.S. Environmental Protection Agency, Department of Energy (DOE), or Oak Ridge Institute of Science and Education (ORISE).

**Acknowledgements**

This work was supported by the U.S. Environmental Protection Agency Office of Research and Development. This research
was supported in part by an appointment to the U.S. Environmental Protection Agency (EPA) Research Participation Program administered by the ORISE through an interagency agreement between the U.S. DOE and the U.S. Environmental Protection Agency. ORISE is managed by ORAU under DOE contract number DE-SC0014664. We thank Kathleen Fahey and Barron Henderson for providing comments on a draft of this manuscript. MMC, RHS, and LX acknowledge support through the EPA-STAR program, Grant # 84001001 and the CIRES cooperative agreement NA17OAR4320101. LX also
acknowledges NASA grant 80NSSC21K1704. EPA does not endorse any products or commercial services mentioned in this publication



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
