# Peer review of "Sensitivity of Northeast U.S. surface ozone predictions to the representation of atmospheric chemistry in CRACMMv1.0"

_EGUsphere, 2023_

## Author Comment (AC1)

Response to reviewers of:
"Sensitivity of Northeast U.S. surface ozone predictions to the representation of atmospheric chemistry in CRACMMv1.0"
Posted on EGUsphere

8 June 2023

Bryan K. Place,[1] William T. Hutzell,[2] K. Wyat Appel,[2] Sara Farrell,[1] Lukas Valin,[2] Benjamin N. Murphy,[2] Karl M. Seltzer,[3] Golam Sarwar,[2] Christine Allen,[4] Ivan R. Piletic,[2] Emma L. D'Ambro,[2] Emily Saunders,[5] Heather Simon,[3] Ana Torres-Vasquez,[1] Jonathan Pleim,[2] Rebecca H. Schwantes,[6] Matthew M. Coggon,[6] Lu Xu,[6,7] William R. Stockwell,[8] and Havala O. T. Pye[2]

[1]Oak Ridge Institute for Science and Engineering (ORISE) Fellow Program at the Office of Research and Development, US Environmental Protection Agency, Research Triangle Park, North Carolina, USA

[2]Office of Research and Development, US Environmental Protection Agency, Research Triangle Park, North Carolina, USA

[3]Office of Air and Radiation, US Environmental Protection Agency, Research Triangle Park, North Carolina, USA

[4]General Dynamics Information Technology, Research Triangle Park, North Carolina, USA

[5]Office of Chemical Safety and Pollution Prevention, US Environmental Protection Agency, Washington D.C, USA

[6]NOAA Chemical Science Laboratory (CSL), Boulder, Colorado, USA

[7]Cooperative Institute for Research in Environmental Science (CIRES), University of Colorado, Boulder, Colorado, USA

[8]University of Texas at El Paso, El Paso, Texas, USA

We would like to thank all the reviewers for taking the time to review and provide feedback on the manuscript. We address all of the reviewer comments in our response below. Reviewer comments are reproduced in black and responses are highlighted in blue. Additions or modifications to the manuscript are underlined and any deleted text is indicated using .

Reviewer #1 comments and responses:

**Comment #1:** This paper evaluates how the newly updated CRACMMv1.0 chemical mechanism does at predicting northeastern US O3 concentrations when compared to older CMAQ mechanisms and surface observations of O3 across the northeast. They find the new mechianism predicts lower concentrations of O3 than the RACM2_ae6 mechanism did, which was closer to surface network observations. Evaluating this mechanism in a box model, they explain these improvements arise (1) largely because of rate constant updates more in line with the state of the science and (2) better representations of aeromatic and monoterpene chemistry. The analysis shown was convicing, well explained, and insightful. Although it is primarily a mechanism evaluation and comparison, the paper also has important policy implications- that controlling aeromatics may be a particularly useful way to reduce urban O3 pollution without rural disadvantages. Overall, I found the paper to be exceptionally well written and of high scientific quality and reccommend accepting it with a minor addition to the code and data availability section.

Response: We thank reviewer #1 for the thoughtful comments and feedback

**Comment #2:** In this section, the authors highlight that the mechanism is available in CMAQ and on github and have pointed users to the F0AM and AMET github sites where users can access the surface O3 data and the box model used in their analysis and state that specific model inputs and code will be archived on data.gov following its final publication. The only minor addition I'm requesting is to make sure the authors intend to include a functioning F0AM mechanism file with the CRACMMv1.0 mechanism they developed in this final data so other F0AM users can utilize this (now field constrained), mechanism to expand the scientific utility of this work. Furthermore, I would encourage the authors to include the SMILES strings for all compounds (and how those relate to their "short names" in the mechanism) as part of this archived data, in order to allow the broader community to use computational tools to compare such a large mechanism to other mechanisms that might lump organic species differently. SMILES codes are provided for the compounds in mechanisms like the MCM and Bates et al., 2022 to enable this exact kind of comparison between different chemical mechanisms. For an example of what I'd like to see to continue this trend of being able to easily compare new chemical mechanisms with many different organic species and lumping schemes, I'd point the authors to the supplement of Bates et al., 2022: https://acp.copernicus.org/articles/22/1467/2022/acp-22-1467-2022-supplement.pdf . For lumped compounds, a few examples of SMARTS strings that would match a lumped compound or a SMARTS string that would return compounds matching that lumping scheme would be sufficient. If these details are added the final data archive, the paper would enable more researchers to easily build off this work and use this new mechanism for other scientific insights.

Response: The CMAQv5.3-5.4 code, AQS model output data from AMET, and F0AM mechanism files used in this study are now posted publicly and can be found at https://doi.org/10.23719/1528552. The F0AM files posted at that doi are also posted on https://github.com/USEPA/CRACMM for easier discoverability. The code availability section has been edited to include the doi:

"Specific analyses and scripts used in this manuscript, such as the modelled and observed ozone concentrations, F0AM box model inputs, and exact CMAQ code used, are archived at: https://doi.org/10.23719/1528552. "

For CRACMM, we decided to link each species to one representative structure for purposes of documentation and property estimation. The SMILES strings for organic compounds are provided in Appendix A of Pye et al. (2023). All compounds (with few exceptions) have SMILES strings in the CMAQ species namelists at https://github.com/USEPA/CMAQ, in a markdown file on https://github.com/USEPA/CRACMM, and archived in the CMAQ code at https://doi.org/10.5281/zenodo.7218076. Exceptions that do not have SMILES strings include: lumped crustal species, coarse sea spray cations, unspeciated anthropogenic particulate matter, primary unspeciated coarse particulate matter, and some tracking species. POC and NCOM are legacy CMAQ species that appear in CRACMM but do not have SMILES representations and are not populated with mass. The code and data availability statement was modified:

"Supporting data for CRACMM including guidance on emission preparation and species metadata (including SMILES identifiers) is available at https://github.com/USEPA/CRACMM (U.S. Environmental Protection Agency, 2022c)."

Reviewer #2 comments and response:

**Comment #1:** This paper presents a new, expanded chemical mechanism for use in the EPA regional model CMAQ. The mechanism is evaluated in CMAQ simulations of the northeast U.S. for summer 2018 through comparison to surface ozone observations and 2 other established chemical mechanisms. The mechanisms are also compared using the F0AM box model to quantify the importance of several groups of VOCs at high and low NOx. The paper is very clearly written and the figures and tables clearly present the results and support the

conclusions that the CRACMM mechanism shows improvements in regions of importance for exposure. I recommend publication after consideration of a few minor points.

Response: We also thank Reviewer #2 for their thoughtful comments and feedback

**Comment #2:** Section 3.2 states some statistics for differences between urban/suburban and rural sites, but these are not easily distinguished in maps or given in a table. It would be helpful to show separate maps of just the urban and rural sites, or distinguish urban vs rural in the current maps with different symbols (circle and star, for example).

Response: Our intention in Section 3.2 was not to explicitly delineate between urban/suburban/rural sites but instead to point out that model/observation biases were typically lower closest to major cities within our domain and higher away from urban centers. In light of this comment we have revised Section 3.2 as follows:

Page 8, lines 225-233:
"Model biases were much lower around the metropolitan NE (Washington, D.C., Maryland, New Jersey, New York City/Long Island regions), where predictions fell within $\pm$ 4 ppb of the observed average values. Linear correlations between hourly $O_3$ estimates and observations at a given AQS site were typically high (r > 0.8) in the Northeast US (Figure 2b). Correlations between hourly observations and predictions were the weakest at sites located in the Appalachian Mountain region (r = 0.4 – 0.6) and were strongest at sites located  in the metropolitan Northeast (r > 0.9). Hourly $O_3$ normalized mean biases (NMB) and normalized mean errors (NME) across the domain can be found in the supplement (Figure S2), and values followed a similar spatial distribution as Figures 2a and 2b with lower NMB (-20% to + 20%) and NME (< 30%) values nearer to  population centers (eg. Washington D. C., Baltimore, Philadelphia, Boston) and higher NMB (+ 20-100%) and NME (> 40%) at sites further from city centers  (Figure S2)."

Page 10, lines 259-265:
"The calculated hourly and MDA8 ozone statistics for the CB6r3_ae7 and RACM2_ae6 simulations were found to be of very similar spatial distribution and magnitude to those calculated for CRACMMv1.0 (Figure 2; Figures S2-S6), where both simulations reported lower biases in the metropolitan NE and higher in other areas of the domain. Given that all mechanism $O_3$ biases were lowest nearer to  major cities, this suggests that the CMAQ simulations better estimated $O_3$ concentrations in areas exposed to higher levels of anthropogenic pollutants."

**Comment #3:** Section 4.1 describes sensitivity simulations where the emissions from different categories of VOCs are set to zero. These cases provide some interesting results and give an indication of the importance of the different sources to ozone production. However, the section should start with a discussion of the challenges of this sort of sensitivity simulation due to the non-linearity of ozone production. It might have been more realistic to perform sensitivity tests with a partial reduction (20%) of emissions.

Response: We would like to thank the reviewer for raising this point. Since this is an initial assessment of the chemical mechanism and running a series of partial reduction emission simulations would have been computationally expensive, we chose to focus solely on the mechanism' responses to complete emission reduction scenarios. This is something that will be taken into consideration as CRACMMv1.0 is continuously tested. In light of this comment we have modified the beginning of Section 4.1 to read as follows:

Page 14, lines 348-352:
"A series of emission sensitivity simulations were performed in CMAQ to gain insight into the precursor ROC systems important for $O_3$ formation in CRACMMv1.0 across the NE US summer 2018 model domain. The sensitivity simulations were conducted by running a set of zero-out simulations (i.e., setting emissions of

a chemical class or emissions sector to zero) and determining the response in $O_3$ concentrations to the emission perturbation. A list of all the emission zero out simulations can be found in Table 2. Due to the non-linear response of ozone production to perturbations in $NO_x$ concentrations the interpretations of zeroed emission simulations can be challenging. Nonetheless, these types of perturbations provide an initial assessment of the ozone production response in CRACMMv1.0 and provide insight into how chemical systems respond to lower $NO_x$ emissions in CRACMMv1.0 versus RACM2_ae6 and CB6r3_ae7."

**Comment #4:** 'zero out' -> 'zero-out' or maybe some other wording in some cases. Such as l.353, which could be written '... a series of zeroed emissions cases.'

Response: All instances of zero-out in the manuscript have been changed to 'zeroed emissions'

**Comment #5:** l.480: CRAMM -> CRACMM

Response: Fixed

**Comment #6:** I am sure the Editors will point out that they do not accept Github for the archiving of code and require a copy to be put on Zenodo.

Response: See response to Reviewer #1 with updates to code availability.

**References in response to reviewers**:

Pye, H. O. T., Place, B. K., Murphy, B. N., Seltzer, K. M., D'Ambro, E. L., Allen, C., Piletic, I. R., Farrell, S., Schwantes, R. H., Coggon, M. M., Saunders, E., Xu, L., Sarwar, G., Hutzell, W. T., Foley, K. M., Pouliot, G., Bash, J., and Stockwell, W. R.: Linking gas, particulate, and toxic endpoints to air emissions in the Community Regional Atmospheric Chemistry Multiphase Mechanism (CRACMM), Atmos. Chem. Phys., 23, 5043–5099, https://doi.org/10.5194/acp-23-5043-2023, 2023.

---

## Author Response (AR2)

A final editor comment was received:

"The ozone performance relies very much on chemical mechanisms but meteorology and boundary conditions from chemicals transported and react to the region are also key too. This could be a further investigation. Perhaps, the performance could be different, especially on different terrains. I suggest authors could add the statement in the text or suggestions."

The role of boundary conditions was previously highlighted. We added one sentence after the one on boundary conditions (line 555) to highlight the role of testing on additional domains:

> ozone was elevated above 50 ppb. Spatially, CRACMMv1.0 showed smaller bias in the Northeast U.S. urban corridor and higher bias at rural sites, particularly in the Appalachian Mountains. Similar results were found for diurnal predictions at individual sites where CRACMM best matched $O_3$ observations at a site that experienced higher $NO_x$ concentrations. As regional boundary conditions for CRACMMv1.0 were obtained from CB6r3_ae7, the full effects of CRACMMv1.0 on
>
> 555 regional background air quality and long range transport predictions have yet to be fully examined. Further, since the coupling of meteorology and chemistry has been shown to play a major role in $O_3$ distributions, the robustness of the mechanism should also be tested on a variety of domains that encompass different terrains.